# Cardio-Metabolic Health of Offspring Exposed in Utero to Human Immuno-Deficiency Virus and Anti-Retroviral Treatment: A Systematic Review

**DOI:** 10.3390/biology13010032

**Published:** 2024-01-06

**Authors:** Edna Ngoakoana Matjuda, Godwill Azeh Engwa, Muhulo Muhau Mungamba, Constance Rufaro Sewani-Rusike, Nandu Goswami, Benedicta Ngwenchi Nkeh-Chungag

**Affiliations:** 1Department of Human Biology, Faculty of Health Sciences, Walter Sisulu University PBX1, Mthatha 5117, South Africa; 217297331@mywsu.ac.za (E.N.M.); mmungamba@wsu.ac.za (M.M.M.); crusike@wsu.ac.za (C.R.S.-R.); 2Department of Biological and Environmental Sciences, Faculty of Health Sciences, Walter Sisulu University PBX1, Mthatha 5117, South Africa; gengwa@wsu.ac.za (G.A.E.); nandu.goswami@medunigraz.at (N.G.); 3Physiology Division, Otto Loewi Research Center for Vascular Biology, Immunology and Inflammation, Medical University of Graz, Neue Stiftingtalstrasse 6, D-5 A, 8036 Graz, Austria; 4Department of Health Sciences, Alma Mater Europaea, 2000 Maribor, Slovenia; 5College of Medicine, Mohammed Bin Rashid University of Medicine and Health Sciences, Dubai P.O. Box 505055, United Arab Emirates

**Keywords:** human immunodeficiency virus, antiretroviral therapy, in utero exposure, foetal environment, cardio-metabolic health, pregnancy, offspring

## Abstract

**Simple Summary:**

Although the use of antiretroviral treatment (ART) is effective in reducing the risk of HIV transmission to the foetus, there are concerns of possible adverse effects of in utero exposure on offspring’s cardio-metabolic health. This paper systematically reviewed the effects of HIV/ART exposure during pregnancy on the cardio-metabolic health of offspring. Reports from 35 eligible studies showed that HIV-exposed uninfected (HEU) children’s cardio-metabolic health was negatively impacted by in utero exposure to ART. A few studies showed direct cardiometabolic risk factors including increased blood pressure and lipids, reduced insulin, oxidative stress, cardiac damage and vascular and myocardial dysfunction among HEU children compared to their HIV-unexposed uninfected (HUU) children while most studies reported indirect cardiovascular risk factors including reduced head circumference, low birth weight, and altered mitochondrial content in HEU children. These findings suggest that in utero exposure of ART may affect foetal health predisposing them to cardiometabolic diseases later in life.

**Abstract:**

Background: Antiretroviral treatment (ART) use during pregnancy continues to rise as it is known to decrease the likelihood of HIV transmission from mother to child. However, it is still unknown whether foetal exposure to (ART) may affect the foetal environment, predisposing the offspring to cardiometabolic risk. Therefore, the aim of this study was to systematically review the cardio-metabolic effects of in utero exposure to HIV/ART on offspring. Methods: We carried out a systematic review and obtained literature from the Google scholar, PubMed, ProQuest, Web of Science, and Scopus databases. Two independent reviewers evaluated the titles, abstracts, and full-length English contents. Data from the eligible studies were included. Results: The search yielded 7596 records. After assessing all of these records, 35 of the full-length articles were included in this systematic review. Several studies showed that low birth weight, small head circumference, and altered mitochondrial content were more common among HIV-exposed uninfected (HEU) children compared to HIV-unexposed uninfected children (HUU). A few studies demonstrated elevated triglyceride levels, lower levels of insulin, and increased blood pressure, oxidative stress, vascular dysfunction, cardiac damage, and myocardial dysfunction among HEU children compared with HUU children. Conclusion: Most findings showed that there were cardio-metabolic health risk factors among HEU children, indicating that maternal exposure to HIV and ART may negatively affect foetal health, which may lead to cardio-metabolic morbidity later in life.

## 1. Introduction

Human immuno-deficiency virus (HIV), which is known to cause acquired immuno-deficiency syndrome (AIDS), continues to be a significant problem with regard to global public health [1]. Over 38 million people worldwide were infected with HIV in 2021. Approximately 54% of these people were women and girls [2]. Before the advent of antiretroviral therapy (ART), women of reproductive age were discouraged from becoming pregnant due to concerns about HIV transmission to their unborn children [3]. However, now that ART is widely available, HIV-positive women of childbearing age can have children with little risk to themselves, their children and partners [4,5]. When used during pregnancy, ART has been reported to be the best intervention to lower the risk of mother-to-child transmission levels to below 5% [6]. This is made evident by the increasing number of HIV-exposed uninfected (HEU) children [7]. In fact, 1.4 million HEU children are born each year [8].

Despite the fact that ART has significantly decreased rates of vertical HIV transmission during pregnancy, mounting evidence is indicating that ART may increase the risk of undesirable pregnancy outcomes, including premature birth and intrauterine growth restriction among children who were exposed to HIV during gestation [9,10]. Intrauterine growth restriction is a known cause of low birth weight and small head size [11]. Additionally, it was reported to be a risk factor for the development of long-term cardiometabolic disease [12]. Further, a study documented that the perinatal environment determines the developmental stages of a foetus and defines their susceptibility to the onset of diseases such as hypertension, diabetes, and metabolic disorders including obesity, dyslipidaemia, insulin resistance, inflammation, and vascular dysfunction, among others, as well as oxidative stress [13,14,15,16]. These undesired pregnancy outcomes affecting offspring after birth may be a result of “foetal programming”, a condition whereby harm to the intrauterine environment results in structural and functional alterations in vital organs in postnatal life, increasing the likelihood of the emergence of numerous diseases in later life [17,18]. 

The intrauterine environment modulates the placenta [19], as it provides the foetus with blood that is rich in nutrition and oxygen during pregnancy. The placenta serves as a conduit for communication between the mother and the foetus. Most toxins and medications cannot pass through the placental barrier, which is made up of chorionic connective tissue, the trophoblastic epithelium covering the villi, and the foetal capillary endothelium. [20]. In some cases, drugs are made to readily cross the placental barrier, as they are beneficial to the foetus. For example, the transplacental passage of ART is known to prevent the perinatal transmission of HIV [21]. However, the chronic administration of ART, often conducted throughout pregnancy, may affect the in utero environment, predisposing the foetus to the risk of developing cardiometabolic diseases after birth. Moreover, maturation and functions of the placenta may be interrupted by persistent exposure to ART [22]. For instance, a study found that ART increases maternal endothelium dysfunction and placental dysfunction, which may increase the risk of pre-eclampsia in pregnant women on ART [23]. Another study carried out in Tygerberg, South Africa, found that children born to HIV-positive women inhabited smaller placentas than children born from healthy women [24]. These results indicate that HIV/ART may have adverse effects on the in utero environment, which may predispose the foetus to future postnatal chronic diseases. The impact of the in utero environment on the health of a child was clearly demonstrated by a Helsinki study conducted on adulthood among men who had low birth weight (LBW). This study found that these men were more susceptible to developing coronary heart disease [25].

Although the use of ART is beneficial to lowering the risk of vertical transmission, it is crucial to investigate any potential negative effects of in utero exposure on offspring. Therefore, the purpose of this study was to systematically review the cardio-metabolic effects of in utero exposure to HIV/ART on offspring born to HIV-infected mothers.

## 2. Methods

### 2.1. Search Strategy

This systematic review adheres to Preferred Reporting Items for Systematic Reviews and Meta-Analyses (PRISMA) guidelines for Systematic Reviews [26]. Online databases, namely, Google scholar, PubMed, ProQuest, and Scopus, were used to conduct a systematic search between March and June 2022. Full-length text case–control studies, longitudinal studies, prospective observational studies, prospective cohort studies, and retrospective observational studies reported in English were included in this systematic review. Studies that included appropriate controls for comparison were used to assess cardiovascular health. The terms “retrospective” and “prospective” describe the timing of research in relation to the acquisition of the results. In retrospective studies, each participant has already experienced the desired outcome (or not, as, for example, in controls) at the time of enrolment, and data are gathered either through records or by asking people to recall exposure. Conversely, in prospective research, individuals are monitored over time to ascertain the occurrence of outcomes because the outcome (and occasionally even the exposure or intervention) has not yet occurred at the initiation of the study [27]. Cross-sectional studies can be completed within a reasonable amount of time depending on the required sample size and accessibility to the study population. The main feature of this type of study design is that the results are generalised for the study population using a cross-sectional sample that is representative of the population [28]. A longitudinal study is the opposite of a cross-sectional study. In longitudinal studies, the same participants are frequently observed over time [29]. In randomized control trials, a homogeneous group of study participants is split into two groups at random. These two groups ought to be identical in every way, including with regard to measured and unmeasured confounders, if the randomization is successful [30].

The Population, Intervention, Comparison, and Outcome (PICO) tool used to develop the present research criteria is displayed in Table 1. The keywords searched for population included pregnant woman, pregnant women, and offspring. The intervention covered drugs used in HIV treatment, HIV, the environment of interest (i.e., in utero), and the period (namely, during pregnancy and the postnatal period). The intervention considered was compared with HIV/ART-free in utero and HIV-uninfected populations. The outcomes were categorised in terms of growth, cardiovascular risk, cardiac effects, and mitochondrial effects. The records resulting from the search were screened manually by two reviewers independently. Disagreements were resolved after a discussion so that extracted data could be merged. Secondary data such as editorials, reviews, and commentaries were not considered. Further, grey literature was not included.

### 2.2. Data Extraction

The study titles were screened, and records of those that were eligible according to the inclusion criteria were exported to Microsoft (MS) Excel. Data that were extracted included the title of a study, its abstract, and its year of publication. Duplicate records and irrelevant studies were excluded, and full-length text articles were selected for this systematic review.

## 3. Results

A total of 7596 records were obtained from the systematic search. Of those records, 7292 were removed for several reasons, including ineligibility and duplication, among others. The remaining 304 articles were assessed based on their titles and abstracts, according to which 238 were excluded. Sixty-six articles were retrieved as full-length studies for comprehensive review. After removing articles based on the outcomes of interest, 35 articles met our eligibility criteria. The details of the study selection process are presented in Figure 1.

Presented below is a PRISMA 2020 flow diagram for new systematic reviews, which included searches of databases and registers only.

The characteristics of the included studies are displayed in Table 2. A few studies reported the onset of ART administration. All the studies published between 2011 and 2022, consisting of original data, and reported in English were included.

Six studies found that infants exposed to HIV/ART in utero had lower birth weights (LBWs) compared to HUU children [31,32,40,48,57,61]. In addition, infants exposed to ART in the first trimester had lower birth weights compared to those exposed later in pregnancy or not at all according to a study carried out in Brazil [36]. Another study carried out in Cape Town, South Africa, reported a higher prevalence of LBW among HEU children (14%) compared to HUU children (9%) [41]. Studies carried out in Uganda and Botswana found that children exposed to HIV in utero were significantly underweight as compared to healthy unexposed, uninfected children [60,66]. In a study carried out in South Africa, the prevalence of underweight was 11.1% for children in the HEU group and 8.4% for children in the HUU group, respectively [57]. However, other studies observed similar birth weights among HEU and HUU newborns [56,58,59,66]. In addition, children in the HEU group in a study conducted in Harare and Gweru, Zimbabwe, had significantly lower weight-for-length z scores than those in the HUU group [37,42]. Another study carried out in South Africa reported that the length-for-age z scores of HEU children were lower compared to those of HUU children at 12 months [38]. 

The head circumferences (HCs) of infants whose mothers were HIV-positive and those whose mothers did not have HIV were similar in Mozambique and Kenya [58,66]. HC was significantly smaller among infants in the HEU group than those in the HUU group in studies conducted in Nigeria, Romania, South Africa, and Uganda [32,35,39,40]. 

Height was significantly greater among HUU children compared to HEU children in Zimbabwe, Romania, and Cameroon [32,40,42]. However, studies conducted in Kenya and Mozambique found that the lengths of HEU children and HUU children were comparable [58,66]. An HIV-positive status for mothers was associated with lower mean birth weight and height of their offspring from birth to 6 months of life in Tanzania [43].

Low birth weight among HEU children was directly associated with the in vivo production of foetal *TNF-α* in a Brazilian study indicating increased pro-inflammatory cytokine levels among neonates born to HIV-positive mothers [51].

A South African study reported that the levels of cytokines such as interferon gamma and interleukin-1 beta (*p* < 0.01) were significantly decreased among HEU infants at 6 to 10 weeks old [34], while peroxidised lipids generated by reactive oxygen species (ROS) were elevated among infants exposed to ART/HIV in Netherlands [44]. This study further reported increased levels of triglycerides among HEU children [44]. A study carried out in Cameroon reported that HEU newborns had lower preprandial insulin levels than HUU neonates at 6 weeks of age [46]. Indirect measurements, including mitochondrial content, were not consistent in studies carried out in Spain. For example, in [49], the authors reported that there was increased mitochondrial content among HIV/ART-exposed children compared to those not exposed, while another study reported decreased mitochondrial content [62]. Elevated adenine–cytosine/thymine–guanine mutations were observed among HEU children compared to HUU infants, though the difference was insignificant (*p* = 0.09) [47]. Elsewhere, a negative relationship between complex IV (CIV) activity levels and mitochondrial DNA (mtDNA) function was observed among HEU children [52]. 

A direct measure such as myocardial wall thickness was reported to be increased among HEU children in studies conducted in Spain [49,50,53]. On the contrary, a study carried out in the United States reported decreased septal wall thickness among HEU infants [54]. Echocardiographic measurements were similar in Spain and in the United States [31,36]. Other studies showed decreased Left-ventricular (LV) mass among HEU children in the United States [45,54]. In addition, LV mass was significantly lower among girls of the same group compared to boys [54], while signs of systolic dysfunction, diastolic dysfunction, and increased cIMT were observed among HEU children [50]. Another study conducted in the United States reported that HEU children had higher heart rates [33]. In addition, a study conducted in the United States reported that HIV-negative children aged 8–12 years exposed to ART in utero had reduced left ventricular diastolic function compared to HUU children of the same age [65]. 

A study conducted in Portugal documented significantly decreased myocardial peak systolic velocities in HIV-exposed children [64]. Significant lower mitral late diastolic inflow velocities and higher adjusted mean LV mass-to-volume-ratio Z-scores were observed among HEU children and adolescents compared to HUU children and adolescents aged 7–16 years old in the United States [55]. A study carried out in Spain reported that the left isovolumic relaxation time was significantly longer in HEU foetuses than that of the HUU foetuses [49], while the foetal ratio between early (E) and late (atrial—A) ventricular filling velocity was significantly higher among HEU children compared to HUU children in Spain [31]. No discernible difference was observed between cardiac damage and perinatal exposure to ART among 3-year-old HEU children and 3-year-old HUU children in a study carried out in the United States [45]. Correlations between cardio biomarkers, LV mass, LV function, and inflammatory markers were observed among HEU youth in the United States [63]. 

## 4. Discussion

This study systematically reviewed the cardio-metabolic health of children exposed to HIV/ART in utero. Most studies were from Africa, where many children are exposed to HIV/ART in utero due to the high prevalence of HIV infection. Despite not being infected with HIV, HEU children are exposed to ART in utero during the crucial phase of their cardiovascular system development [35,55]. In this review, some studies were focused on LBW, alter mitochondrial content and HC as indirect cardiovascular risk factors, while others assessed the direct cardio-metabolic risk factors, including insulin resistance, lipid disorders, inflammation, oxidative stress, vascular dysfunction, cardiac damage and cardiac dysfunction following in utero exposure to HIV and ART. Selected cardiovascular disease markers were assessed in some studies, while mitochondrial DNA changes relating to cardiac health were assessed in others.

Pregnancy causes physiological changes that differently affect how all drugs are absorbed, distributed, metabolised, and eliminated in pregnant compared to non-pregnant women [67]. Even though the primary goal of drug therapy during pregnancy is to treat maternal problems, dose regimens are nearly always decided without taking into account the major modifications of drug handling by the pregnant woman. Furthermore, the majority of drugs cross the placenta easily, exposing the foetus to treatments. In most global guidelines and clinical settings, nucleoside reverse transcriptase inhibitors (NRTIs) are known to be the backbone of ART for pregnant women [67,68]. They prevent the virus from replicating itself. However, exposure to NRTI is known to be associated with mitochondrial toxicity [69]. For example, a study conducted in Spain found that the mitochondrial content of HEU children was significantly increased [49]. Loss of mitochondrial function, which was observed in HEU children exposed to ART in utero [47,49,52,62], may lead to an increase in the release of lipids into circulation, where they reduce nitric oxide, which is an important vasodilator that acts by indirectly mediating oxidative stress and downregulating insulin signalling [70,71]. Increased concentrations of insulin were observed in HEU children compared with those of HUU children [46]. In a study conducted in the Netherlands, the authors reported increased triglyceride levels as well as production of ROS-catalysed peroxidation metabolites among HEU infants [44]. This suggests that in utero exposure to ART may affect mitochondrial function, which may lead to dyslipidaemia, hyperglycaemia, and oxidative stress, affecting their metabolic health and posing a possible risk of CVD events in future [46].

HEU children exposed to NRTI in utero had increased carotid intima–media thickness (cIMT) and systolic blood pressure (SBP) and diastolic blood pressure (DBP) dysfunction, which may indicate a higher cardiovascular risk in later life for HIV-negative infants exposed to ART during pregnancy [50]. Significant elevated levels of circulating inflammatory cytokines were observed among HEU neonates in Brazil and may be associated with foetal inflammatory response syndrome. There is a need for the long-term observation of these infants in order to ascertain the impact of ART administered during pregnancy on the cellular immune system’s response to different antigens in later life [51]. A study in the United States linked increased inflammatory biomarkers with left-ventricular (LV) mass, LV function, and LV wall stress in the corresponding HEU group. This suggests inflammation may affect the heart function of the exposed children, increasing their risk for CVDs. Exposure to highly active antiretroviral therapy (HAART) was positively correlated with LV fractional shortening among perinatal HIV-infected children when compared with HIV-infected children without HAART exposure [72]. In addition, a study documented that by late adolescence, pulse wave velocity (PWV) was higher in perinatal HIV-infected children than in their HIV-negative peers, and PWV was positively correlated with higher arterial pressure, a CVD risk factor [73]. Another study carried out in the Unted States, Australia, and Europe involving HIV patients documented that protease inhibitors were associated with myocardial infarction [74]. These studies suggest that inflammation, high blood pressure, and vascular dysfunction may result in cardiac damage and dysfunction, increasing the risk for CVDs. Exposure to ART during pregnancy has been linked to minor direct cardiac effects [55]. For example, studies carried out in Spain revealed that HEU foetuses exposed to Zidovudine exhibited cardiac remodelling [49,50,53]. This finding could account for cardiovascular alterations in childhood [53]. The LV dimension and myocardial peak velocities were decreased among HEU children in the United States and Portugal, respectively [54,64]. These changes suggest an overall loss of cardiac tissue associated with ART exposure and may lead to progressive systolic dysfunction later in life [54,64].

Some South African studies reported decreased weight among HEU children [38,41]. This finding is in agreement with studies conducted in other African countries such as Tanzania, Nigeria, and Zimbabwe [32,37,43]. This finding suggests that both the actual maternal viral infection and the use of ART could directly affect foetal growth [40,75]. Furthermore, children whose mothers started ART before conception birthed infants with higher preterm births compared to those who started ART after pregnancy, which might have led to low birth weight [57]. However, other studies, including those conducted in South Africa, Malawi, Uganda, Botswana, Cameroon, and Mozambique, reported similar weights among HEU children and HUU children [34,56,60,66]. Length was also not consistent in Africa. For instance, low length was reported among HEU children in South African studies as well as in Zimbabwe and Uganda [35,37,38,39], whereas other studies carried out in South Africa, Botswana, and Mozambique documented similar results between children in the HEU group and HUU group [34,60,66]. Moreover, a study reported that intrauterine growth restriction increased the amount of nutrients supplied to the nervous system at the expense of other important systems. As a result, this restriction may impact cardiovascular and postnatal metabolism later in life, as reported in a study on post-menopausal women, which revealed that their LBW was linked to a higher risk of CVD in their adulthood [76]. A study carried out in South Africa reported that preterm delivery appeared to be more common among women who conceived on ART compared to those initiating treatment during pregnancy, which may explain the low birth weight among these children [41]. Further, underweight among children was associated with maternal CD4 [56]. Rapid weight increase in the early postnatal years is frequently associated with restricted foetal growth that is reflected by LBW, which exacerbates harmful health effects, including obesity. [77]. HIV-positive individuals are more likely to be obese, and abnormalities in fat distribution persist even with modern ART [69]. However, children not exposed to HIV but exposed to nurturing environments have been shown to exhibit equally dramatic gains in length and weight among adolescents in Romania [78]. 

It is known that head circumference at birth reflects intrauterine growth, which could lead to further cardiovascular events in the future. A study documented that having a small head size at birth increases the risk of death from coronary heart disease (CHD) [79]. Another study reported that smaller head circumference at delivery as a result of restricted foetal development was linked to an elevated likelihood of cardiovascular death [18]. Further, in [80], the authors reported hypoplastic left heart syndrome, which is a cardiovascular malformation affecting infants born with small heads. ART has also been shown to be associated with the head size of children after birth. Studies carried out in Cameroon, Nigeria, South Africa, and Uganda reported significantly lower head circumferences among children in the HIV/ART in utero exposure group compared to children in the HUU group [32,35,39,40]. In addition, this study showed a robust relationship between small head circumference and efavirenz-based ART. Further, this study also showed protective effects of darunavir with respect to head size [81]. This finding suggests that different types of ART could have different effects on the physical growth of children, including head size, rendering them prone to cardiovascular risk in the future. Therefore, changes in the body’s structure, physiology, and metabolism as a result of ART exposure throughout the foetal phase could be the origin of CVD [18].

One of this review’s shortcomings is that the included studies did not focus on the in utero effect of specific ART drugs. Only a few studies reported the effect of individual classes of ART in HEU children. Thus, it is not clear whether the observed cardiovascular risk observed for children in the HEU group is related to the specific ART treatment or class. The cardiovascular effect observed could not be established based on the duration of the administration of ART for pregnant women, as most studies reported that women started ART at different times. This might have influenced the quality of the results reported. 

## 5. Conclusions

In utero exposure to HIV/ART had an adverse effect on the cardio-metabolic health of HIV-exposed uninfected children. Although there were sufficient data on indirect cardiovascular risk factors including intrauterine growth restriction, low birth weight, head circumference, and altered mitochondrial content, there were few data on direct cardio-metabolic biomarkers including dyslipidaemia, insulin resistance, hypertension, oxidative stress, vascular and myocardial dysfunction and cardiac damage, which are major cardiovascular disease risk factors. Therefore, more studies, especially those focusing on direct cardiovascular risk factors are required to investigate and monitor the cardiometabolic health of children exposed to HIV and ART in utero.

## Figures and Tables

**Figure 1 biology-13-00032-f001:**
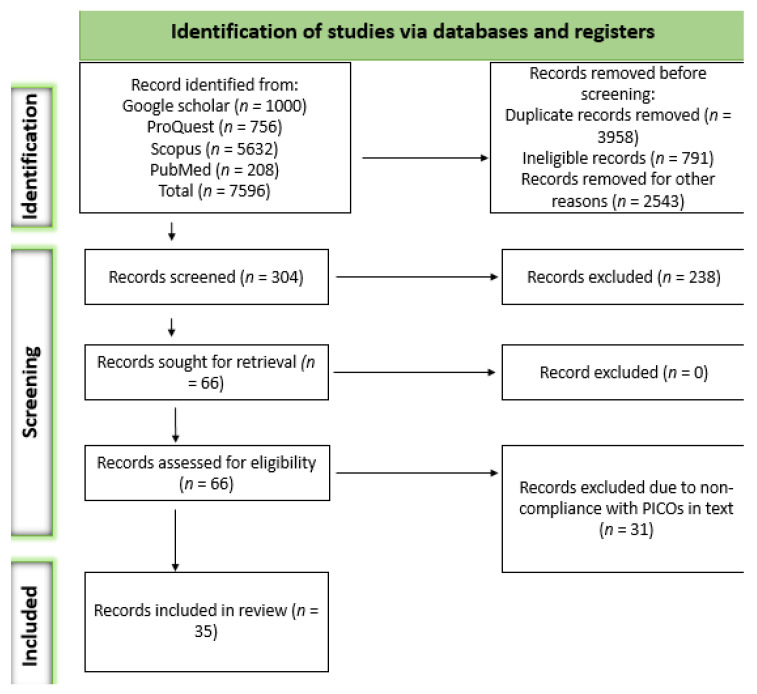
Search and screening selection process based on PRISMA 2020 guidelines.

**Table 1 biology-13-00032-t001:** Search criteria.

Category	Specific Category	Keywords	Search Number	Search Mask
Population	Inclusion criteria	“pregnant woman” OR “pregnant women” OR “maternal” OR “Foetus” OR “Fetus” OR “new-born” OR “neonates” OR “infant” OR “child” OR “children” OR “offspring”	1	All fields
Intervention	Drugs	“ART” OR “antiretroviral therapy” OR “HAART” OR “highly active antiretroviral therapy” OR “anti-HIV agents” OR “HIV drug”	2	All fields
	Environment	“In-utero” OR “intrauterine” OR ”foetal” OR “fetal”	3	All fields
	Infection	“HIV positive” OR “human immunodeficiency virus” or “HIV positive” OR “HIV” OR HIV OR “HIV infection”	4	All fields
	Period	“pregnancy” OR “gestation” OR “prenatal” OR “postnatal” OR “perinatal”	5	
Comparison	Study group	“HIV exposed uninfected” (HEU) children or HIV-positive pregnant women	6	All fields
	Control group	“HIV unexposed uninfected” (HUU) children or HIV negative pregnant women	7	
Outcome	Anthropometry	“weight” OR “low birth weight” OR “LBW” OR “high birth weight” OR “HBW” OR “weight for age” OR “WAZ” OR “height” OR “length” OR “length for age” OR “LAZ” OR “head circumference” OR “HC”	8	All fields
	Cardiac effects	“cardiac biometry” OR “cardiac toxicity” OR “diastole” OR “diastolic” OR “systolic” OR “systole”	9	All fields
	Cardiovascular risk factors	“obesity” OR “dyslipidaemia” OR “hypertension” OR “high blood pressure” OR “inflammation” OR “inflammatory” OR “oxidative stress” OR “reactive oxygen species” OR “ROS” or “diabetes” OR “vascular dysfunction” OR “endothelial dysfunction” OR “lipid”	10	All fields
	Mitochondrial effects	“mitochondrial toxicity” OR “mitochondrial DNA” OR “mtDNA” OR “mutation”	11	All fields
	Excluded terms	“COVID-19” OR “animal” OR “simian immunodeficiency virus” OR “SIV” OR “SARS-CoV 2” OR “coronavirus” OR “nervous system disease” OR “malaria” OR “cancer” OR “tumor” OR “malignant” OR “malignance”	12	

**Table 2 biology-13-00032-t002:** Characteristics of the included studies.

Country	Study Design	Sample Size	Onset of ART Treatment	ART	Outcome	Indirect/Direct Measure	Citation
Spain	Longitudinal cohort	99 mother–infant pairs	10 women before gestation; 19 women during gestation	Mothers = 2NRTI (3TC, d4T, FTC); ZDV; 1NNRTI (NVP)/1 PI (NFV, RTV, LPV) ATV. HEU children = ZDV	Increased rates of low-birth-weight infants in the HEU group Foetal cardiac biometric variables were similar in both groups The foetal E/A tricuspid was higher in the group exposed to HIV	Indirect and direct	[31]
Nigeria	Longitudinal cohort	131 HUU and 141 HEU infants	N/A	N/A	Birthweight was lower among HEU children	Indirect	[32]
United States	Longitudinal cohort	148 HEU and 130 HUU children	N/A	N/A	Heart rates were higher in HEU children. The overall mitral valve peak early-diastolic velocity (E-wave) was lower in the HEU cohort	Direct	[33]
South Africa	Longitudinal cohort	77 HEU and 190 HUU children	41.6% before pregnancy; 58.4% during pregnancy	N/A	Weight, length, and head circumference at 6 weeks and 2 years old were similar Levels of pro-inflammatory cytokines such as *IL*-1β and IFN-γ were reduced in HEU infants at 6 to 10 weeks and 24 to 28 weeks	Direct and indirect	[34]
South Africa	Observational prospective pilot	22 HUU infants; 32 HEU infants	N/A	Mothers = TDF, FTC, EFV, AZT, 3TC, LPV	Smaller head circumferences at birth among HEU infants	Indirect	[35]
Brazil	Prospective cohort	155 HUU and 433 HEU infants	114 infants in the 1st trimester; 319 infants in the 2nd or 3rd trimester	HEU children = ZDV; cotrimoxazole after 6 weeks old	Lower birth weight among HEU infants	Indirect	[36]
Zimbabwe	Prospective cohort	52 HEU and 55 HUU children	N/A	Neonates = NVP; administration of cotrimoxazole at 6 weeks old.	Lower weight-for-length z scores among HEU children	Indirect	[37]
South Africa	Prospective cohort	461 HEU and 411 HUU children	HIV-positive women during 1st antenatal clinic visit	N/A	The weight-for-age z score of HEU infants aged 9 months old was lower At 12 months, HEU group had lower length-for-age z scores	Indirect	[38]
Malawi and Uganda	Prospective cohort	471 HEU and 462 HUU children	N/A	Women = ZDV; cART regimens.	At 24 months, HEU children’s head circumferences were smaller At 12 and 24 months, Ugandan HEU children’s length-for-age Z scores were lower	Indirect	[39]
Cameroon	Prospective cohort	3737 mother–infant pairs	89.3% of pregnant women before or during pregnancy	N/A	HEU children’s birth weight and head circumference were smaller The prevalence of small for gestational age (SGA) was higher among HEU infants	Indirect	[40]
South Africa	Prospective cohort	299 HIV-women and 1494 HIV+ women	922 women during pregnancy; 572 women before conceiving	87% women = (TDF, 3TC EFV); 4% (TDF, 3TC and NVP; 3% (PI); 6% (other NRTI)	LBW was noted in neonates born to HIV-positive mothers.	Indirect	[41]
Zimbabwe	Prospective cohort	3120 HEU and 9210 HUU children	N/A	HEU infants = co-trimoxazole	Lower length-for-age Z-scores and weight-for-length Z-scores, as well as greater rates of stunting and wasting, among HEU infants Lower length for age among the HEU infants from birth to 24 months	Indirect	[42]
Tanzania	Prospective cohort	70 HIV-women and 44 HIV+ women	15 women before pregnancy. 24 women during pregnancy.	Women = EFV, AZT, 3TC (CD4 < 350 cells/μL). AZT (other HIV+ women).	Lower mean newborn weight and length among HEU infants. Decreased WAZ at 2, 3, and 6 months, as well as lower LAZ at 2 months, among HEU infants	Indirect	[43]
Netherlands	Prospective cohort	12 HEU and 15 HUU infants	5 women before pregnancy	Mothers = 2NRTI (ZDV, 3TC); either a PI (RTV-boosted LPV or NFV) or an NNRTI (NVP)	Increased production of ROS catalysed lipid peroxidation metabolites among HEU infants Levels of triglycerides and diacylglycerols were higher in HEU newborns	Direct	[44]
United States	Prospective cohort	417 HEU and 98 HUU children	N/A	Mothers = NRTI, NNRTI, and PI	Echocardiographic Z-scores did not significantly differ between the HEU and HUU children	Direct	[45]
Cameroon	Prospective cohort	210 HUU and 156 HEU children	N/A	Women = cART (CD4 < 350 cells/μL); Other women and infants = AZT + NVP	Smaller head circumference and increased glucose-to-insulin ratio among HEU children	Direct and indirect	[46]
Columbia	Prospective cohort	57 HEU and 70 HUU infants	12 women before conception; 45 women during pregnancy	Women = ZDV/3TC/d4T/dddI/ABC/FTC/TDF	Adenine–cytocine/thymine–Guannine mutations were more common in HEU children.	Direct	[47]
South Africa	Prospective cohort	431 HEU and 457 HUU infants	62% of women before conception; 32% after conception	N/A	HEU infants showed lower WAZ and LBW values.	Indirect	[48]
Spain	Prospective cohort	47 HIV+; 47-pregnant women	78.7% of women before pregnancy	Women = NRTI; either a NNRTI/1PI/INI	Large left myocardial wall thickness, increased left isovolumic relaxation time, and increased mitochondrial content among the HEU foetuses	Direct	[49]
Spain	Prospective cohort	34 HEU and 53 HUU infants	79.4% before pregnancy	Women = 2NRTI; 1NNRTI/1 boosted PI or 1 INI	Thicker myocardial septal wall and cIMT among HEU infants Increased SBP and DBP among HEU group	Direct	[50]
Brazil	Prospective cohort	12 HIV− pregnant; 80 HIV+ pregnant women	20 women at delivery. 60 women at 20–32 weeks gestation	Women = 2 NRTIs (AZT,3TC/ddI, 3TC) with PI(LPV/SQV)	Increased *TNF-α* and IL1B levels low birth weight among ART-exposed neonates	Direct and indirect	[51]
Spain	Prospective observational	133 HEU and 73 HUU infants	111 women at 2–40 weeks of gestation.	Women = NRTI, NVP NFV, ZDV (mothers)	Reduced mitochondrial function in HIV/ARV-exposed healthy children,	Indirect	[52]
Spain	Prospective cohort	42 HIV+; 84 HIV− pregnant women	32 women before pregnancy.	Women = 2NRTI; 1PI	larger hearts and pericardial effusion together with thicker myocardial septal walls in HEU foetus. systolic and diastolic dysfunction in HEU foetus. Maternal treatment with zidovudine was the only factor associated with foetal cardiac changes	Direct	[53]
United States	Prospective multisite cohort	136 HEU HIV+ infants	N/A	N/A	LV mass was lower among HEU girls from birth to 2 years of age Septal wall thickness and LV dimension were lower among HEU infants	Direct	[54]
United States	Prospective cohort	156 HEU and 18 HUU children and adolescents	N/A	Women = 87 Non-HAART ARV; 67 HAART	Increased lower mitral late diastolic inflow velocities and LV mass-to-volume ratio Z-scores for HEU children and adolescents	Direct	[55]
Uganda	Retrospective cohort	1380 HUU and 122 HEU children	5 HIV mothers during gestation	Women = 1NVP; cotrimoxazole	The birth weights of HEU and HUU newborns were comparable.	Indirect	[56]
South Africa	Cross-sectional survey	6179 HUU 2599 HEU newborns,	N/A	Women = TDF; 3TC/FTC, NVP if CD4 ≤ 350 cell/µL or ZDV from 14 weeks. Infants = NVP	Newborns in the HEU group had higher rates of SGA, LBW, and being underweight.	Indirect	[57]
Cameroon	Pilot cross-sectional	102 mother-neonate pairs	N/A	N/A	Similar LBW	Indirect	[58]
Kenya	Cross-sectional survey	277 mother- infant pairs	63% of women during pregnancy	Women = TDF/ AZT/3TC/NVP; d24/3TC/NVP TDF/3TC/NVP/TDF/3TC/EFV	Slight variations in weight and WAZ between infants were associated with TDF in mothers	Indirect	[59]
Botswana	Comparative cross-sectional	154 HEU and 259 HUU children	N/A	N/A	The infants’ weights and lengths in the HEU and HUU groups were comparable.	Indirect	[60]
Spain	Cross-sectional	32 HIV− t; 24 HIV+ pregnant women	84% of pregnant women before pregnancy	NRTI, 2NRTI, 1PI, 2NRTI/, 2PI, 2NRTI, /NNRTI/AZT monotherapy (mothers)	Lower birth weight among HEU children	Indirect and direct	[61]
Spain	Cross-sectional, controlled observational	35 HIV− women; 27 HIV+ pregnant women	4 women in 2nd stage of pregnancy; 23 women before gestation	2NRTI, 1PI/NRTI, 1NNRTI	Lower mitochondrial content in HEU infants; SGA was observed among HEU children.	Indirect and direct	[62]
United States	Cross-sectional analysis	156 HEU and HIV youth	N/A	Women = HAART, PI (159); HAART (54), non-HAART ART (17)	Increased inflammatory markers, increased LV wall mass and stress, and lower LV function among HEU youths	Direct	[63]
Portugal	Case control	77 HEU children; 38 HUU children	N/A	N/A	Lower myocardial peak systolic velocities in HEU children.	Direct	[64]
United States	Cross-sectional	30 HEU and 30 HUU children	N/A	Women = 10% ZTV; 90% cART	Reduced left ventricular diastolic function	Direct	[65]
Mozambique	Randomized control trial	561 HIV+ and 1183 HIV− pregnant women	21% of women at delivery	Women = AZT; HEU infants = NVP	Birth weight, HC, and length at birth were similar between HEU and HUU children.	Indirect	[66]

HEU = HIV-exposed and -uninfected, HUU = HIV-unexposed and -uninfected, SGA = small for gestational age, LBW = low birth weight, LAZ = length-for-age z score, *IL*-1β = Interleukin-1Beta, IFN-γFNy = Interferon-y, ROS = Reactive oxygen species, cART = Combined antiretroviral therapy, N/A = Not applicable, WAZ = weight-for-age z-score, HC = head circumference, CIV = complex IV, NRTI = Nucleoside reverse transcriptase inhibitor, 3TC = Lamivudine, d4T = Stavudine, FTC = Emtricitabine, ZDV= Zidovudine, NNRTI = Non-nucleoside reverse transcriptase inhibitor, NVP = Nevirapine, PI = Protease inhibitor, NFV = Nelfinavir, RTV = Ritonavir, LPV = Lopinavir, ATV = Atazanavir, ZDV = Zidovudine, HIV+ = human-immunodeficiency-virus-positive, HIV− = human-immunodeficiency-virus-negative, TDF = Tenofovir, EFV = Efavirenz, ABC = Abacavir, ddl = didanosine, INI = Intergrase inhibitor, HAART = Highly active antiretroviral therapy, ARV = Anti-retroviral, cell/µL = cells per microliter, ART= antiretroviral treatment, LV = Left-ventricular, *TNF-α* = Tumour necrosis factor, cIMT = carotid intima media thickness, SBP = Systolic blood pressure, and DBP = Diastolic blood pressure.

## Data Availability

Not applicable.

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
