# Peer review of "Cardio-Metabolic Health of Offspring Exposed in Utero to Human Immuno-Deficiency Virus and Anti-Retroviral Treatment: A Systematic Review"

_biology, 2024, doi:10.3390/biology13010032_

Round 1

Reviewer 1 Report

Comments and Suggestions for Authors

This is a timely review on the potential effects of antiretroviral therapy on the developing cardiac system. While the authors present an extensive methodology on how articles were chosen, it seems too restrictive and misses very important literature in an already limited field. The discussion could be enhanced to provide new insights into what is known about HEU and cardiac function. Listed below are specific concerns:

1. Provide the societal impact of HEU, there is no mention of the estimated 1.4 million HEU children born each year. There are many HEU children now into their adolescent and early adult years, what is the known about their cardiometabolic health? See Garcia-Otero et al., 2019 which was completely missed in this review. The Guerra et al., paper should be more prominent in this discussion. Mellin et al., 2022 could also be used, but not cited in the current manuscript.

2. Although specific clinical studies examining individual AR drugs is limited, a discussion on the potential effects of ART drugs should be presented along with the pharmacokinetic data on most widely used drugs and their ability to cross the placental barrier. This could then tie in with the discussion about the little that is known and need for further investigation. The authors could even lean on adult data to draw the potential link between specific ARTs on heart function.

3. As stated earlier, there are a handful of articles that for some reason were excluded from this article that should at least be revisited: 

P. Martins et al., Myocardial peak systolic velocity-a tool for cardiac screening of HIV-exposed uninfected children. Eur J Pediatr 179, 395-404 (2020).

L. García-Otero et al., Cardiac and mitochondrial function in HIV-uninfected fetuses exposed to antiretroviral treatment. PLoS One 14, e0213279 (2019).

L. García-Otero et al., Cardiac Remodeling and Hypertension in HIV-Uninfected Infants Exposed in utero to Antiretroviral Therapy. Clin Infect Dis 73, 586-593 (2021).

J. D. Wilkinson et al., Cardiac and inflammatory biomarkers in perinatally HIV-infected and HIV-exposed uninfected children. AIDS 32, 1267-1277 (2018).

T. M. Kasahara et al., The impact of maternal anti-retroviral therapy on cytokine profile in the uninfected neonates. Hum Immunol 74, 1051-1056 (2013).

A. Noguera-Julian et al., Decreased Mitochondrial Function Among Healthy Infants Exposed to Antiretrovirals During Gestation, Delivery and the Neonatal Period. Pediatr Infect Dis J 34, 1349-1354 (2015).

S. W. Worm et al., Risk of myocardial infarction in patients with HIV infection exposed to specific individual antiretroviral drugs from the 3 major drug classes: the data collection on adverse events of anti-HIV drugs (D:A:D) study. J Infect Dis 201, 318-330 (2010).

L. García-Otero et al., Zidovudine treatment in HIV-infected pregnant women is associated with fetal cardiac remodelling. AIDS 30, 1393-1401 (2016).

S. E. Lipshultz et al., Cardiac Effects of Highly Active Antiretroviral Therapy in Perinatally HIV-Infected Children: The CHAART-2 Study. J Am Coll Cardiol 70, 2240-2247 (2017).

S. Dirajlal-Fargo, G. A. McComsey, Cardiometabolic Complications in Youth With Perinatally Acquired HIV in the Era of Antiretroviral Therapy. Curr HIV/AIDS Rep 18, 424-435 (2021).

S. E. Lipshultz et al., Cardiac effects of antiretroviral therapy in HIV-negative infants born to HIV-positive mothers: NHLBI CHAART-1 (National Heart, Lung, and Blood Institute Cardiovascular Status of HAART Therapy in HIV-Exposed Infants and Children cohort study). J Am Coll Cardiol 57, 76-85 (2011).

Comments on the Quality of English Language

Language is acceptable.

Author Response

 The Editor,

MDPI Biology,

Dear Sir,

The authors are grateful to the reviewers for taking their time to assess our manuscript by giving critical review and constructive criticism of the manuscript 2700697 titled “Cardio-metabolic health of offspring exposed in-utero to human immune-deficiency virus and anti-retroviral treatment: A systematic review”. Below is our response to reviewers’ comments. All corrections in the updated manuscript are highlighted in red colour.

  1. Provide the societal impact of HEU,
    1. there is no mention of the estimated 1.4 million HEU children born each year.

Response: Thank you for your suggestion, This has been added in the introduction. Please see line 37-38

“ In fact, 1.4 million HEU children are born each year [8].”

  1. There are many HEU children now into their adolescent and early adult years, what is the known about their cardiometabolic health?

Response: Information of HEU children into their adolescent and early adult yearshas been added. Please see lines 147-149,150-152 and 156 to 159 in the results section

“In addition, a study conducted in the United States reported that HIV negative children aged 8-12 years exposed to ART in-utero had reduced left ventricular diastolic function compared to HIV unexposed uninfected (HUU) children of the same age [65].

Significant lower mitral late diastolic inflow velocities and higher adjusted mean LV mass-to-volume ratio Z-scores were observed among HEU children and adolescents compared to HUU children and adolescents aged 7-16 years old in the United States [55]. A study carried out in Spain reported that the left isovolumic relaxation time was significantly longer in HEU foetuses than that of the HUU foetuses [49]. The correlations between cardio biomarkers, LV mass, LV function and inflammatory markers, were observed among HEU youth in the United States [63]. A study conducted in Brazil reported significantly increased pro-inflammatory cytokines among neonates born to HIV positive mothers [51].”

  1. See Garcia-Otero et al., 2019 which was completely missed in this review

Response: Thank you for your suggestion. This has been added. Please see Table 2 with citation [49]

Please see lines 136-138

“Indirect measure including mitochondrial content was not consistent in studies carried out in Spain. For example, a study conducted by [49], documented increased mitochondrial content among HIV/ART exposed children compared to those not exposed.”

Lines 142-143

“A direct measure such as myocardial wall thickness was reported to be increased among HEU children in studies conducted in Spain [49, 50, 53]. ”

Lines 152-154

“A study carried out in Spain reported that the left isovolumic relaxation time was significantly longer in HEU foetus than that of the HUU foetus [49].”

  1. The Guerra et al., paper should be more prominent in this discussion.

Response: Thank you for your suggestion. We have included the referred articles. Please see Table 2 with citation [55] and lines 151-153

“Significant lower mitral late diastolic inflow velocities and higher adjusted mean LV mass-to-volume ratio Z-scores were observed among HEU children and adolescents compared to HUU children and adolescents aged 7-16 years old in the United States [55].

Lines 164-165

“Despite not being infected with HIV, HEU children are exposed to in-utero ART during the crucial phase of their cardiovascular system development [55; 35].”

  1. Mellin et al., 2022 could also be used, but not cited in the current manuscript-

Please see lines 191-193

 “In addition, a study documented that by late adolescence, pulse wave velocity (PWV) was higher in perinatal HIV than in HIV-negative peers, and PWV was positively correlated with higher arterial pressure, a CVD risk factor [73].”

  1. Although specific clinical studies examining individual AR drugs is limited, a discussion on the potential effects of ART drugs should be presented along with the pharmacokinetic data on most widely used drugs and their ability to cross the placental barrier. This could then tie in with the discussion about the little that is known and need for further investigation. The authors could even lean on adult data to draw the potential link between specific ARTs on heart function.

This has been added.

Response: Thank you for your response. This information has been added in the discussion section. Please see lines 169-175

“Pregnancy causes physiological changes that affect how all drugs are absorbed, distributed, metabolised, and eliminated differently in pregnant than in non-pregnant women [67]. Even though the primary goal of drug therapy during pregnancy is to treat maternal problems, dose regimens are nearly always decided without taking in account the major modifications of drug handling by the pregnant woman. Furthermore, the majority of drugs cross the placenta easily, exposing the foetus to treatments. In most global guidelines and clinical settings, nucleoside reverse transcriptase inhibitors (NRTIs) are known to be the backbone of ART for pregnant women [67, 68]. They prevent the virus from replicating itself. However, exposure to NRTI is known to be associated with mitochondrial toxicity [69].’’

Lines 196-197

“Exposure to ART during pregnancy has been linked to minor direct cardiac effects [55]. For example, studies carried out in Spain revealed that HEU foetuses exposed to Zidovudine had cardiac remodeling [49, 50, 53].”

Lines 193-194

Another study carried out in the Unted States, Australia and Europe among HIV patients documented that protease inhibitors were associated with myocardial infarction [74].

  1. As stated earlier, there are a handful of articles that for some reason were excluded from this article that should at least be revisited: 
  2. Martins et al., Myocardial peak systolic velocity-a tool for cardiac screening of HIV-exposed uninfected children. Eur J Pediatr 179, 395-404 (2020).

Response: This has been added. Please Table 2 with citation [64]

Please see lines 150-151

“A study conducted in Portugal documented significantly decreased myocardial peak systolic velocities in HIV exposed children [64].”

See lines 198-200

“The LV dimension and myocardial peak velocities were decreased among HEU children in the United States and in Portugal, respectively [54, 64]. These changes suggest an overall loss of cardiac tissue with ART exposure and may lead to progressive systolic dysfunction later in life [54, 64].”

  1. García-Otero et al., Cardiac and mitochondrial function in HIV-uninfected fetuses exposed to antiretroviral treatment. PLoS One 14, e0213279 (2019).

Response: Thank you for your suggestion. This has been added. Please see Table 2 with citation [49]

Please see lines 136-138

“Indirect measure including mitochondrial content was not consistent in studies carried out in Spain. For example, a study conducted by [49], documented increased mitochondrial content among HIV/ART exposed children compared to those not exposed..”

Lines 142-143

“A direct measure such as myocardial wall thickness was reported to be increased among HEU children in studies conducted in Spain [49, 50, 53].”

Lines 143-144

“A study carried out in Spain reported that the left isovolumic relaxation time was significantly longer in HEU foetuses than that of the HUU foetuses [49].”

  1. García-Otero et al., Cardiac Remodeling and Hypertension in HIV-Uninfected Infants Exposed in utero to Antiretroviral Therapy. Clin Infect Dis 73, 586-593 (2021).

Response: Thank you for your suggestion. Please see Table 2 with citation [50]

Please see lines 142-143

“A direct measure such as myocardial wall thickness was reported to be increased among HEU children in studies conducted in Spain [49, 50, 53].”

Line 146-147

“While signs of systolic dysfunction, diastolic dysfunction and increased cIMT among HEU children [50]”

Lines 196-197

For example, studies carried out in Spain revealed that HEU fetuses exposed to Zidovudine had cardiac remodeling [49, 50, 53].

  1. D. Wilkinson et al., Cardiac and inflammatory biomarkers in perinatally HIV-infected and HIV-exposed uninfected children. AIDS 32, 1267-1277 (2018).

Response: Please see Table 2 with citation [63]

Lines 156-158

The correlations between cardio biomarkers, LV mass, LV function and inflammatory markers, were observed among HEU youth in the United States [63].

  1. M. Kasahara et al., The impact of maternal anti-retroviral therapy on cytokine profile in the uninfected neonates. Hum Immunol 74, 1051-1056 (2013).

Response: This has been added. Please see Table 2 with citation [51]

Please see line 131

“Low birth weight among HEU children was directly associated with in-vivo production of foetal TNF-α [51].”

Lines 186-188

“There is a need for long-term observation of these infants in order to ascertain the impact of ART during pregnancy on the cellular immune system's response to different antigens in later life [51].”

Noguera-Julian et al., Decreased Mitochondrial Function Among Healthy Infants Exposed to Antiretrovirals During Gestation, Delivery and the Neonatal Period. Pediatr Infect Dis J 34, 1349-1354 (2015).

Response: Please see Table 2 with citation [52]

Please see lines 139-141

“While a negative relationship between CIV activity levels and mtDNA function was observed among HEU children [52]. 

  1. W. Worm et al., Risk of myocardial infarction in patients with HIV infection exposed to specific individual antiretroviral drugs from the 3 major drug classes: the data collection on adverse events of anti-HIV drugs (D:A:D) study. J Infect Dis 201, 318-330 (2010).

Response: We did not include this study in the results since it assessed two HIV infected groups with and without myocardial infarction which is not our target group. However, it was added in the discussion section.

Please see lines 194-195

“Another study carried out in the Unted States, Australia and Europe among HIV patients documented that protease inhibitors were associated with myocardial infarction [74]. “

  1. García-Otero et al., Zidovudine treatment in HIV-infected pregnant women is associated with fetal cardiac remodelling. AIDS 30, 1393-1401 (2016).

Response: This has been added. Please see Table 2 with citation 53

Please see lines 142-143

“A direct measure such as myocardial wall thickness was reported to be increased among HEU children in studies conducted in Spain [49, 50, 53].”

Lines 196-198

“Exposure to ART during pregnancy has been linked to minor direct cardiac effects [55]. For example, studies carried out in Spain revealed that HEU foetuses exposed to Zidovudine had cardiac remodeling [49, 50, 53]. This finding could account for cardiovascular alterations in childhood [53].”

  1. E. Lipshultz et al., Cardiac Effects of Highly Active Antiretroviral Therapy in Perinatally HIV-Infected Children: The CHAART-2 Study. J Am Coll Cardiol 70, 2240-2247 (2017).

 Response: We did not include this study in the results section as it investigatedHIV infected  childrenexposed or not-exposedto ART which is not our target group. However, the study was included in the discussion section.

Please see lines 190-191

 “Exposure to HAART was positively correlated with left ventricular (LV) fractional shortening among perinatal HIV infected children as opposed to HIV infected without HAART exposure [72].”

  1. Dirajlal-Fargo, G. A. McComsey, Cardiometabolic Complications in Youth With Perinatally Acquired HIV in the Era of Antiretroviral Therapy. Curr HIV/AIDS Rep 18, 424-435 (2021).

Response: We Did not include this study in the results section since it is a systematic review. However, it was added in the discussion section.

Please see line 168-169

 “Exposure to NRTI is known to be associated with mitochondrial toxicity [70]”

Lines 216-217

“HIV-positive individuals are more likely to be obese and abnormalities in fat distribution persist even with modern ART [69]”

Lines 174-175

“They prevent the virus from replicating itself. However, exposure to NRTI is known to be associated with mitochondrial toxicity [69].”

  1. E. Lipshultz et al., Cardiac effects of antiretroviral therapy in HIV-negative infants born to HIV-positive mothers: NHLBI CHAART-1 (National Heart, Lung, and Blood Institute Cardiovascular Status of HAART Therapy in HIV-Exposed Infants and Children cohort study). J Am Coll Cardiol 57, 76-85 (2011).

Response: This study has been added. See Table 2 with citation [54]

Please see lines 143-146

“On the contrary, a study carried out in the United States reported decreased septal wall thickness among HEU infants [54].  Echocardiographic measures were similar in Spain and in the United States [31, 36]. Other studies showed decreased LV mass among HEU children the United States [ 45, 54]. In addition, LV mass was significantly smaller among girls of the same group compared to boys [54]..”

Lines 198-200

“The LV dimension and myocardial peak velocities were decreased among HEU children in the United States and in Portugal, respectively [54, 64]. These changes suggest an overall loss of cardiac tissue with ART exposure and may lead to progressive systolic dysfunction later in life [54, 64].”

Sincerely,

Benedicta Nkeh-Chungag, on behalf of the authors

Reviewer 2 Report

Comments and Suggestions for Authors

This review summarizes findings on cardiometabolic effects in offsprings exposed in utero to HIV/ART and born to HIV positive mothers. The authors have given a decent amount of introduction for studying cardiometabolic effects in such newborns and have presented their selection criteria of literature published in this area in a reasonable manner. However, some key information such as specific drugs used in ART for each cited study is missing and such information can be presented as multiple tables in the review. 

Please find below my specific comments: 

1.     Rearrange table to sort all of the different cited studies into prospective, longitudinal, or cross-sectional and include a brief description of what each of these terminologies indicate in methods

2.     Recommend to add in a separate column for specific ART drugs used in the cited study

3.     Add a column to indicate whether readouts used in each of the cited study are direct or indirect measures of cardiometabolic effects

4.     Indicate which of the measures are consistent across different regions and/or ethnicities: such as low birth weight effect in HEU vs that in HUU is consistent across studies from Brazil and South Africa.

5.     For a given measure, there are differences observed in readouts, as an example- low birth weight in a given cohort but not from other ethnicities or regions. The authors discuss combination of drugs used in ART regimen and duration of onset of ART regimen as potential reasons for these divergent effects; however, discussion or additional information on viral load, inflammatory status of HIV+ positive mothers on ART, any pattern in readouts indicated categorize into those from Western countries such as USA (where there is high prevalence of obesity and other metabolic disorders) vs African countries (which are more prone to infectious diseases) will enhance the value of this study.

6.     Also, recommend the authors to describe phenotypes in HEU vs HUU rather than this way for some parameters and in HUU vs HEU for others for clarity and consistency throughout the review. 

Comments on the Quality of English Language

Overall, the review is well written with sufficient background, body and discussion of presented findings. However, recommend the authors to review specific sentences as indicated in the attached pdf file to clarify a finding or conclusion. 

Author Response

The Editor,

MDPI Biology,

Dear Sir,

The authors are grateful to the reviewers for taking their time to assess our manuscript by giving critical review and constructive criticism of the manuscript 2700697 titled “Cardio-metabolic health of offspring exposed in-utero to human immune-deficiency virus and anti-retroviral treatment: A systematic review”. Below is our response to reviewers’ comments. All corrections in the updated manuscript are highlighted in red colour.

Please find below my specific comments: 

  1. Rearrange table to sort all of the different cited studies into prospective, longitudinal, or cross-sectional

Response: Thank you for your suggestion; This has been done. Please see Table 2. Study design column

  1. Include a brief description of what each of these terminologies indicate in methods

Response: This has been included in the methods section

Please see lines 71-81

“The terms “retrospective” and “prospective” describes the timing of research in relation to the development of the results. In retrospective studies, each participant has already experienced the desired outcome (or not—for example, in controls) at the time of enrollment and data are gathered either through records or by asking people to recall exposure. Conversely, in prospective research, individuals are monitored over time to ascertain the occurrence of outcomes because the outcome (and occasionally even the exposure or intervention) has not yet occurred at the time the study begins [27]. Cross-sectional study can be finished within a reasonable amount of time depending on required sample size and accessibility to study population. The main feature of this type of study design is that the results are generalised for the study population using a cross-sectional sample that is representative of the population [28]. A longitudinal study is the opposite of a cross-sectional study. In longitudinal studies, same participants are frequently observed over time [29]. In randomized control trials, a homogeneous group of study participants is split into two groups at random. These two groups ought to be identical in every way, including measured and unmeasured confounders, if the randomization is successful [30]. “

  1. Recommend to add in a separate column for specific ART drugs used in the cited study

Response: This has been added. Please see ART column in Table 2

  1. Add a column to indicate whether readouts used in each of the cited study are direct or indirect measures of cardiometabolic effects

Response: This has been added. Please see Direct/Indirect Measures in Table 2

  1. Indcate which of the measures are consistent across different regions and/or ethnicities: such as low birth weight effect in HEU vs that in HUU is consistent across studies from Brazil and South Africa.

Response: This information has been added in the results and discussion sections.

Please see lines 136-139

“Indirect measure including mitochondrial content was not consistent in studies carried out in Spain. For example, a study conducted by [49], documented increased mitochondrial content among HIV/ART exposed children compared to those not exposed. While another study reported decreased mitochondrial content [62].”

Lines 142-144

“A direct measure such as myocardial wall thickness was reported to be increased among HEU children in studies conducted in Spain [49, 50, 53]. On the contrary, a study carried out in the United States reported decreased septal wall thickness among HEU infants [54].  “

Lines 201-202

“Some South African studies reported decreased weight among HEU children [38, 41]. This was in agreement with studies conducted in other African countries such as, Tanzania, Nigeria, Zimbabwe [37, 32, 43]..”

Lines 205-209

“other studies including those conducted in South Africa, Malawi, Uganda, Botswana, Cameroon and Mozambique reported similar weights among HEU children and HUU children [34, 56, 60, 66]. Length was also not consistent in Africa. For instance, low length was reported among HEU children in South African studies, Zimbabwe and Uganda [35, 37,38, 39]. whereas other studies carried out in South Africa, Botswana and Mozambique documented similar results between children in the HEU group and HUU group [34, 60, 66].”

Lines 224-225

“Studies carried out in Cameroon, Nigeria, South Africa and Uganda reported significantly lower head circumference among children in the HIV/ART in-utero exposure as contrasted to children in the HUU group [40,32, 35,39].”

  1. For a given measure, there are differences observed in readouts, as an example- low birth weight in a given cohort but not from other ethnicities or regions. The authors discuss combination of drugs used in ART regimen and duration of onset of ART regimen as potential reasons for these divergent effects; however, discussion or additional information on viral load, inflammatory status of HIV+ positive mothers on ART, any pattern in readouts indicated categorize into those from Western countries such as USA (where there is high prevalence of obesity and other metabolic disorders) vs African countries (which are more prone to infectious diseases) will enhance the value of this study.

Response: Very few studies included viral load, inflammatory marker status or other makers of HIV+ mothers and thus it was difficult to classify these readouts according to ethnicity or region. Only a few did. Please see lines 212-214

“A study carried out in South Africa reported that preterm delivery appeared to be more common among women conceiving on ART as compared to those initiating during pregnancy which may explain low birth weight among these children [41]. Further, underweight in children was associated with maternal CD4 [56]. “

  1. Also, recommend the authors to describe phenotypes in HEU vs HUU rather than this way for some parameters and in HUU vs HEU for others for clarity and consistency throughout the review.

Response: Only a few studies groups HEU and HUU children by sex. See lines 139-140

“In addition, LV mass was significantly smaller among girls of the same group compared to boys [54].”

  1. peer-review-32975080.v1.pdf Font size selected is different

Response: This has been fixed.

  1. Reframe the sentence for clarity

Response: This has been fixed. See line 51-54

In some cases, drugs are made to readily cross the placental barrier as they are beneficial to the foetus. For example, the transplacental passage of ART is known to prevent perinatal transmission of HIV [21]. However, chronic administration of ART, often throughout pregnancy may affect the in-utero environment predisposing the foetus to risk of cardiometabolic diseases after birth.

  1. Please check this for accuracy - how do you explain higher levels of inflammatory markers in infants born to HIV+ mothers who themselves have lower levels of serum inflammatory markers?

Response: This has been Fixed. Please lines 131-133

A South African study reported that cytokines such as interferon gamma and interleukin-1 beta (p<0.01) were significantly decreased among HEU children infants at 6 to 10 weeks old [34]..

  1. Which country it was reported in?

Response: Please see line 135-136

 A study carried out in Cameroon reported that HEU newborns had lower preprandial insulin levels than HUU neonates at 6 weeks of age [46].

  1. Do the authors mean more direct measures here?

Response: the statement has been revised as shown in line 165-168

“In this review, some studies were focused on LBW and HC as indirect cardiovascular risk factors while others assessed the direct cardiovascular risk factors including insulin resistance, lipid disorders, inflammation, oxidative stress, vascular dysfunction and cardiac dysfunction following in-utero exposure to HIV and ART”

  1. Please check this sentence for clarity. Post-menopausal women referred to here are they mothers or kids born to HIV+ mothers upon hitting menopause are at high risk for CVD?

Response: the article did not provide information whether the post-menopausal women were HIV positive or not. However, the important information is that their low birth weight was associated with the cardiovascular events in adulthood, implying LBW may lead to future cardiovascular events after birth. The statement has been revised in the discussion section as thus ¨ As a result, this restriction may impact cardiovascular and postnatal metabolism later in life as reported by a study on post-menopausal women which revealed that their LBW was linked to a higher risk of CVD in their adulthood [76]. (see lines 210-212)

  1. Once again please specify are the authors referring to HEU kids here?

Response: They are not HEU children. The stament has been revised for clarity. See lines 216 to 218

HIV-positive individuals are more likely to be obese and abnormalities in fat distribution persist even with modern ART [69]. Although children not exposed to HIV but exposed to nurturing environments have equally been shown with dramatic gains in length and weight among adolescents in Romania [78].

  1. Please specify women mentioned here are mothers or girl child born to HIV+ mothers who in adulthood are at risk for CHD?

Response: This study is not HIV positive. It is to support the hypothesis that small head size may lead to cardiovascular diseases.

See line 220

A study documented that having a small head size at birth increases the risk of death from coronary heart disease (CHD) [79].

Sincerely,

Benedicta Nkeh-Chungag, on behalf of the authors

Round 2

Reviewer 2 Report

Comments and Suggestions for Authors

The authors have addressed my comments and have included additional requested information that will be useful to the readers and enhance the impact of the review. 

A minor typo in vivo in line 131needs to be italicized.